# TransLinkGuard: Safeguarding Transformer Models Against Model Stealing in Edge Deployment

## ABSTRACT

Proprietary large language models (LLMs) have been widely applied in various scenarios. Additionally, deploying LLMs on edge devices is trending for efficiency and privacy reasons. However, edge deployment of proprietary LLMs introduces new security challenges: edge-deployed models are exposed as white-box accessible to users, enabling adversaries to conduct effective model stealing (MS) attacks. Unfortunately, existing defense mechanisms fail to provide effective protection. Specifically, we identify four critical protection properties that existing methods fail to simultaneously satisfy: (1) maintaining protection after a model is physically copied; (2) authorizing model access at request level; (3) safeguarding runtime reverse engineering; (4) achieving high security with negligible runtime overhead. To address the above issues, we propose TransLinkGuard, a plug-and-play model protection approach against model stealing on edge devices. The core part of TransLinkGuard is a lightweight authorization module residing in a secure environment, e.g., TEE. The authorization module can freshly authorize each request based on its input. Extensive experiments show that TransLinkGuard achieves the same security protection as the black-box security guarantees with negligible overhead.

## CCS CONCEPTS

• **Security and privacy** → **Social aspects of security and privacy**; **Authorization**; • **Computing methodologies** → *Natural language processing*.

## KEYWORDS

Intellectual Property Protection, Edge-deployed Transformer Model, Authorization, Trusted Execution Environment

## 1 INTRODUCTION

Large language models (LLMs), especially proprietary LLMs, such as ChatGPT [14], Gemini [3], and Claude [39], have achieved astounding success in recent years, demonstrating remarkable capabilities on myriad tasks [4, 43]. Typically, interaction with these proprietary models purely relies on APIs (Figure (1a)), where users submit prompts and receive outputs from API (referred to as API-based access) [40]. However, due to concerns about user privacy, high bandwidth costs, and latency inherent in this API-based access, the edge deployment of LLMs has emerged as an alternative [30]. This

Permission to make digital or hard copies of all or part of this work for personal or classroom use is granted without fee provided that copies are not made or distributed for profit or commercial advantage and that copies bear this notice and the full citation on the first page. Copyrights for components of this work owned by others than the author(s) must be honored. Abstracting with credit is permitted. To copy otherwise, or republish, to post on servers or to redistribute to lists, requires prior specific permission and/or a fee. Request permissions from permissions@acm.org.
*ACM MM, 2024, Melbourne, Australia*
© 2024 Copyright held by the owner/author(s). Publication rights licensed to ACM.
ACM ISBN 978-x-xxxx-xxxx-x/YY/MM
https://doi.org/10.1145/nnnnnnn.nnnnnnn

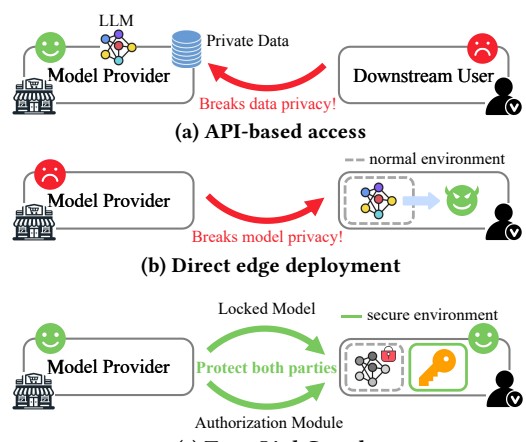

**Figure 1: Paradigms of interacting with LLMs. (a) API-based access: users send data to the model owner. (b) Direct edge deployment: the model is straightforwardly deployed in a normal environment. (c) TransLinkGuard: deploy the locked models in a normal environment and the corresponding authorization module in a secure environment.**

approach addresses these concerns by keeping model interaction within the environment that the users have full access to.

However, straightforward edge deployment of proprietary LLMs also introduces new security threats to the deployed LLMs (Figure (1b)): by making models white-box accessible to users, adversaries can obtain full model information (including architecture and weights) and easily achieve high attack effectiveness of model stealing (MS) [21, 41, 42]. Given the significant investment required to develop high-performance LLMs [53], it is essential to protect the intellectual property of the models produced by providers. Therefore, one key objective of the edge deployment of LLMs is to protect these deployed models. Ideally, the protection can downgrade such white-box (with whole model information) MS attacks to **black-box** settings (with only model query access).

Unfortunately, as shown in Table 1, traditional solutions struggle to protect the intellectual property of edge-deployed models as they fail to address the diverse requirements. Specifically, passive protection methods, such as watermark [1, 22, 31], are not applicable since only the proof of ownership is insufficient in such an unsupervised edge operation scenario, where attackers can misuse the model without detection. In contrast, active authorization protection works by allowing only authorized users to use the well-performed model [12, 15, 65]. For example, only users who possess the key can use the model, thereby achieving authorization (referred to as key-based access). However, these methods provide only a model-level authorization. Specifically, once authorization is completed (i.e., the key is distributed), anyone can copy and misuse

| Solutions (exemplar) | Proactivity | Request-level authorization | Runtime security | **Security** with **efficiency** |
|---|---|---|---|---|
| Watermarking [1] | ✗ | ✗ | ✓ | ✓ |
| Key-based access [15] | ✓ | ✗ | ✗ | ✓ |
| Model encryption [65] | ✓ | ✗ | ✗ | ✓ |
| PTSE [36] | ✓ | ✓ | ✓ | ✗ |
| TransLinkGuard (ours) | ✓ | ✓ | ✓ | ✓ |

**Table 1: Comparison with existing solutions. ✓/✗ illustrates whether the method can achieve the corresponding property.**

the model with the key. To avoid the model being copied and mis-used, some work encrypts it before deploying it on devices [33, 65], and these models are only decrypted before execution. However, it's crucial to recognize that while these solutions can implement effective access control before the inference state, current studies [5, 59] suggest that, even after authorization, models remain susceptible to runtime attacks during inference, i.e., attackers reverse engineer the model in its runtime state.

To defend against runtime attacks, one potential solution [6, 37] is to place the model into a secure execution environment, e.g., a trusted execution environment (TEE). TEE is an isolated hardware enclave that stores sensitive data and safeguards against runtime attacks. However, straightforward black-box protection by TEEs is impractical because shielding entire LLMs within TEEs results in a roughly 50× reduction in model efficiency due to TEEs' limited computational speed [55]. Thus, some researchers propose only putting a subset of the model in TEEs and offloading the rest of the computation to GPUs, i.e., Partial TEE-Shielded Execution (PTSE) [36, 49, 51]. Nonetheless, TEE's poor computational power still causes PTSE solutions to struggle with balancing security and efficiency (proved in recent studies [64]). Specifically, due to efficiency demands, the computations that can be executed within the TEE are extremely limited, compelling PTSE to offload a significant number of layers to GPUs. This constraint opens a vulnerability: attackers can replicate the majority of the model offloaded to GPUs and, with minimal training, restore the protected segments, i.e., achieve MS attacks successfully.

Considering the limitations of existing defense strategies, we identify four challenges (**C**) to the intellectual property protection of edge-deployed LLMs. **C1**: Achieving proactive protection to ensure the deployed model remains unusable even if it is physically obtained by attackers. **C2**: Continuously protecting the model beyond model-level authorization, i.e., demanding request-level authorization for every access. **C3**: Ensuring the protection remains effective against runtime attacks. **C4**: Ensuring security while minimizing model runtime overhead.

To ensure the security of edge-deployed LLMs, we propose a plug-and-play transformer model protection approach, TransLinkGuard (Figure (1c)), which addresses all the aforementioned challenges. Specifically, to address **C1**, TransLinkGuard deploys a locked model as a substitute for the original model. The locked model is designed to function normally only when correct authorization is granted for each request by an authorization module (addressing **C2**). Therefore, even if attackers obtain the locked model, it cannot be used without the authorization module. Given the importance of securing this authorization module, it is placed in a secure environment, e.g.,

TEE (addressing **C3**). In this framework, model owners can enforce request-level access control of the edge-deployed model through TEE.

The key challenge in implementing TransLinkGuard is to achieve the authorization mechanism that fulfills the lightweight requirement, i.e., addressing **C4**. To this end, we propose a permutation strategy that row-permutes the weights matrix of linear layers within the model, ensuring that only the corresponding column-permuted input can correctly be computed with the permuted layers. Therefore, as a prerequisite, the input features of the permuted layers must be authorized by an authorization module, which adjusts the features according to the permutation order of this layer before they can be processed by the permuted layer. Consequently, the authorization module requires minimal overhead, as it merely involves rearranging feature elements. Conversely, unauthorized users, lacking knowledge of the permutation order, cannot effectively utilize the permuted layer, even if they can obtain all its parameters. This lightweight nature ensures that even if the authorization mechanism is deployed to all transformer layers, its overhead remains negligible. That is, TransLinkGuard still guarantees efficiency under sufficient security.

Our evaluation shows that TransLinkGuard outperforms existing PTSE approaches in terms of security guarantee and efficiency cost. Attackers can hardly obtain any performance promotion by MS compared to the black-box baseline (i.e., shielding the whole model in TEE). Besides, the experiment, consistent with formulaic derivation, shows no change between the accuracy of the TransLinkGuard-protected model and the original model. The contributions of this work are as follows:

- We systematically identify the requirements for intellectual property protection of edge-deployed LLMs: proactivity, request-level authorization, runtime security, and efficiency. We propose TransLinkGuard, a plug-and-play solution that can protect the edge-deployed transformer models with all these requirements fulfilled.
- TransLinkGuard utilizes a permutation strategy to achieve request-level authorization for edge-deployed LLMs. Compatible with the limited computational speed of TEEs, the lightweight nature of the authorization module surmounts the restriction of PTSE solutions and ensures protection across all transformer layers, thereby enhancing security.
- Extensive experiments demonstrate that compared to the existing PTSE approaches, our proposed TransLinkGuard offers a higher security guarantee with lower overhead and no accuracy loss.

## 2 BACKGROUND AND PROBLEM STATEMENT

### 2.1 Background

**TEE**. A Trusted Execution Environment (TEE) is an isolated hardware enclave that stores and processes sensitive data. Popular TEE implementations include Intel SGX [34], AMD SEV [25], and TrustZone [2]. In this paper, we follow prior work and deem TEE a secure area on a potential adversary host device (including GPUs). This means the data, code, and computation processes inside TEEs are secure. Although there are side-channel attacks that may leak sensitive data from TEE, they are out of the scope of this paper.

**PTSE Sloutions**. Partial TEE-Shielded Execution (PTSE) solutions aim to provide protection against MS by shielding and executing partial models inside TEEs. The motivation of existing work is to reduce inference latency of the straightforward black-box protection that shields the whole model inside TEEs (latency up to 50× [55]). Beyond efficiency considerations, the security goal of PTSE solutions is to downgrade white-box MS against edge-deployed models to black-box attacks. Such degeneration is important for edge-deployed models in the LLMs supply chain.

**One Time Pad**. The One Time Pad (OTP) represents the pinnacle of encryption [48], providing unparalleled secrecy by encrypting messages with a key as long as the message itself. The strength of the OTP lies in its simplicity and the fact that decryption is impossible without the key [47]. In this paper, we leverage the OTP to enhance the security of the authorization process.

### 2.2 Threat Model

In this paper, we consider two parties: the defender and the attacker. The defender is the party that owns the model deployed on an edge device. The attacker attempts to steal the model from the device. We explore a more realistic edge deployment scenario in which the defender attempts to deploy a customized task-specific model aligned with user needs. This makes defense more challenging, as attackers familiar with the task (e.g., possessing some datasets) could facilitate model stealing. The following are the details of the two parts.

**Defender's Goal**. The primary goal of the defender is to ensure the deployed model (donate as $M_{vic}$) only works when proper authorization is given by the trusted hardware (i.e., TEE) within the device. To ensure efficiency, the defender offloads most of the computations to a GPU, which can be accessed in a white-box manner by the user. In the context of MS attacks, the defender's goal is to degrade white-box attacks to black-box settings (the attackers cannot access $M_{vic}$'s weights).

**Adversary's Capability**. To obtain a model with similar performance of authorized $M_{vic}$, the attacker attempts to develop a surrogate model $M_{sur}$ (can work independently) that mirrors $M_{vic}$'s performance on customized tasks. The attacker inspects the offloaded part of $M_{vic}$ in a white-box manner to improve the effectiveness of MS. Specifically, the attacker can infer the architecture of the whole protected model based on the offloaded part with the existing techniques [7, 8] and obtain all the weights in the offloaded part of $M_{vic}$. Besides, we assume that the attacker possesses some well-labeled datasets (less than 1% of the training data) of the task, a practical assumption shared by prior work [19, 45, 60, 64].

### 2.3 Model Stealing Attack

We consider the MS attack, which can obtain the $M_{sur}$, as the security benchmark for protection approaches. Following the prior work [64], we leverage the attack implementation specifically designed for edge-deployed models. Specifically, for an edge-deployed $M_{vic}$ (comprising both protected and offloaded parts), attackers may exploit the offloaded parts to enhance the effectiveness of their attacks.

**Attack Pipeline**. The attack pipeline consists of two phases: *surrogate model initialization* ($P_1$), and *parameter reconstruction* ($P_2$). In $P_1$, the attack begins by inferring the architecture of $M_{vic}$ through its offloaded parts and outputs with existing techniques [7, 8]. Following this, an initial surrogate model, $M_{init}$, is constructed with the same architecture as $M_{vic}$. Finally, the attacker transports $M_{vic}$'s offloaded weights to the corresponding parts of $M_{init}$. In $P_2$, the attacker attempts to replicate the functionality of $M_{vic}$ on $M_{init}$. To this end, one potential approach is to train $M_{init}$ with the dataset they possess to recover the backbone, thus outputting $M_{sur}$. In this process, we consider a more commonly used and effective training method, namely full-parameter training.

## 3 DESIGN OF TRANSLINKGUARD

We argue that the fundamental weakness of PTSE solutions is that all PTSE approaches follow a direct execution strategy, which crudely loads parameter computation into the TEE for protection [64]. The TEE's limitation on speed restricts PTSE from protecting only a small portion of parameters, thereby leading to security vulnerability. With this regard, we champion that an ideal solution should avoid direct model computation execution by the TEE while protecting the model parameters.

We propose TransLinkGuard, a model intellectual property protection approach that protects models through a permutation strategy rather than direct execution. Specifically, TransLinkGuard, tailored for transformer models, protects every linear layer in models through weight permutation. This permutation swaps the positions of each row within the weight matrix of the linear layer. In this way, the positional information of the weights is disrupted, preventing unauthorized users (who are unaware of the permutation order) from utilizing the permuted layers, thus achieving access control. However, with the knowledge of permutation order, the input feature can be column permuted correspondingly so that the elements within the feature correspond positionally with the permuted weights, thus enabling authorized usage. Given that the permutation matrix is crucial for authorization, its protection is essential. To ensure its security, TransLinkGuard secures the authorization process within a TEE to provide a hardware-level guarantee. Furthermore, on the algorithmic level, TransLinkGuard is inspired by previous work [20, 64] and introduces a One-Time Pad (OTP) to encrypt the authorization process.

### 3.1 Approach Overview

This section presents TransLinkGuard, fulfilling requirements in Table 1. Given the subtle structural differences among various transformer models, as the most common scenario, we demonstrate its application on the most classic transformer structure [56].

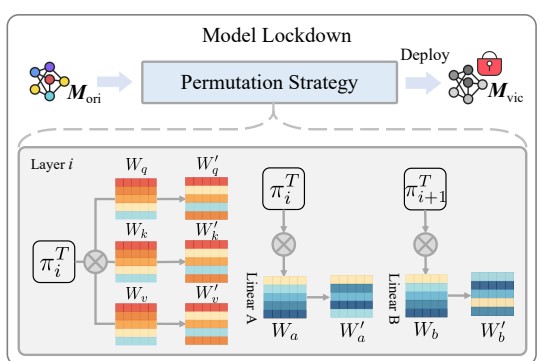
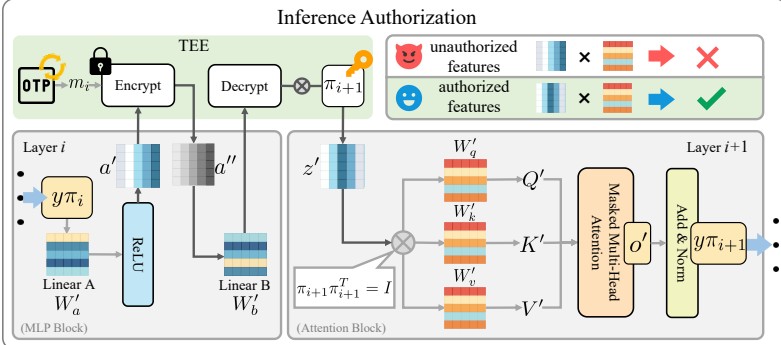

**Figure 2: An overview of TransLinkGuard. (a) Model lockdown: TransLinkGuard uses permutation matrices to permute each transformer layer in $M_{ori}$, creating a locked model $M_{vic}$. (b) Inference authorization: as a prerequisite, the input features of the permuted layers must be authorized before they can be processed by the permuted layer. To facilitate this, the authorization process is integrated within the MLP block of the preceding transformer layer.**

As shown in figure 2, our proposed TransLinkGuard operates in two phases: model lockdown (before deployment) and inference authorization (after deployment). In the model lockdown phase, taking the pre-trained $M_{ori}$ as input, TransLinkGuard randomly initializes different confidential permutation matrices for each transformer layer. To generate the locked model $M_{vic}$, each transformer layer is permuted according to its respective permutation matrix. Therefore, for a permuted transformer layer, its input feature must be authorized correspondingly to achieve accurate computation.

In the inference authorization phase, the authorization module is integrated within the MLP block of the preceding transformer layer, ensuring that features are authorized before they enter the permuted transformer layer. This authorization module takes features as input and outputs the authorized features. To enhance the security, we integrate the authorization mechanism with a linear layer, which involves more parameters and thus makes the authorization process difficult to crack. Considering the limited capacity of TEEs for such a large number of parameter computations, we offload this linear layer to a GPU. Furthermore, To ensure the security of feature transmission between the GPU and TEE during this process, we employ OTP to encrypt the features. Consequently, the authorization process is divided into two steps. Specifically, the first step takes place after the ReLU layer in the MLP block, where the TEE encrypts the feature using OTP. The encrypted feature then undergoes dense linear operations (the second linear layer of the MLP block) on the GPU. In the second step, the feature is decrypted and permuted to complete the authorization.

### 3.2 Model Lockdown

Given a transformer layer (consisting of an attention block and an MLP block), we introduce how to permute its weights (specifically within the attention block) for protection. Although the MLP block is also protected, we will introduce it in section 3.3 as it necessitates integration with the authorization module.

***Attention Block Formalization***. Let $x \in \mathbb{R}^{l \times d}$ denote the input where $l$ is the sequence length (e.g., the number of tokens) and $d$ is the model dimension. We define an attention block as a function $f_\theta : \mathbb{R}^{l \times d} \to \mathbb{R}^{l \times d}$ with weight parameters $\theta$. Then the attention block (including the attention mechanism and its subsequent normalization layer), i.e., $f_\theta(x) = y$, is computed as follows:

$$Q = xW_q, K = xW_k, V = xW_v, \qquad W_q, W_k, W_v \in \mathbb{R}^{d \times d},$$
$$o = \text{softmax}\left(\frac{QK^T}{\sqrt{k}} + M\right)VW_o, \qquad M \in \mathbb{R}^{n \times n}, W_o \in \mathbb{R}^{d \times d}, \quad (1)$$
$$y = \gamma_1 \odot \frac{o + x - \mu_{o+x}}{\sigma_{o+x}} + \beta_1, \qquad \gamma_1, \beta_1 \in \mathbb{R}^d,$$

where $k$ is a constant equal to $d$ divided by the number of attention heads, $M$ denotes the mask, which is an all-zero matrix in the encoder and a matrix whose upper right corner is negative infinity in the decoder. The parameter $\theta$ consists of attention weights ($W_q$, $W_k$, $W_v$, $W_o$), LayerNorm weights ($\gamma_1$, $\beta_1$).

***Permutation Protocol***. Let $\pi_i \in \{0, 1\}^{d \times d}$ denote a permutation matrix of the $i$-th attention block, where $\forall \pi_i, \pi_i \pi_i^T = I$, with $I$ is identity matrix, a property characteristic of permutation matrix. We permute the parameters $\theta$ as follows:

$$W_q' = \pi_i^T W_q, W_k' = \pi_i^T W_k, W_v' = \pi_i^T W_v,$$
$$W_o' = W_o \pi_i, \gamma_1' = \gamma_1 \pi_i, \beta_1' = \beta_1 \pi_i. \quad (2)$$

With the permuted parameters (denoted as $\theta'$), $f_{\theta'}(x\pi_i)$ can be described as follows :

$$Q' = x\pi_i \pi_i^T W_q = xW_q = Q,$$
$$K' = x\pi_i \pi_i^T W_k = xW_k = K,$$
$$V' = x\pi_i \pi_i^T W_v = xW_v = V,$$
$$o' = \text{softmax}\left(\frac{QK^T}{\sqrt{k}} + M\right)VW_o\pi_i = o\pi_i, \quad (3)$$
$$y' = \gamma_1\pi_i \odot \frac{o\pi_i + x\pi_i - \mu_{x+o}}{\sigma_{x+o}} + \beta_1\pi_i = y\pi_i.$$

The functionality of the permuted attention block can be represented as $f_{\theta'}(x') = y\pi_i = f_\theta(x)\pi_i$, valid only when $x' = x\pi_i$.

### 3.3 Inference Authorization

The authorization module design addresses functionality and security. Given that $f_{\theta'}(x')$ requires permuted input for accurate computation, the authorization process, tied to the MLP module,

ensures this prerequisite is met (i.e., permutes the feature by $\pi$). Furthermore, the security of the authorization module is ensured by encrypting features by OTP and involving more model parameters. The authorization process is summarized in Algorithm 1.

---

**Algorithm 1** Algorithm for authorization protocol

---

**Require:** $y\pi_i, W'_a, W'_b, \gamma'_2, \beta'_2, \pi_{i+1}, m$
**Ensure:** $z\pi_{i+1}$
  1: Calculate with the first linear layer as Eq.(6).      // in GPU
  2: Encrypt feature by the one-time mask as Eq.(7).    // in TEE
  3: Calculate with the second linear as Eq.(8).        // in GPU
  4: Decrypt feature and permutation as Eq.(9).      // in TEE
  5: Permutate $y\pi_i$ to $y\pi_{i+1}$ as Eq.(9).            // in TEE
  6: Get $z'$ ($z\pi_{i+1}$) as Eq.(9).                    // in TEE
    **return** $z\pi_{i+1}$

---

***MLP Block Formalization.*** Considering a classic MLP module that receives $y$ from the prior attention block as input, we define its function $g_w : \mathbb{R}^{l\times d} \to \mathbb{R}^{l\times d}$ with weights $w$. This block (including layer norm), i.e., $g_w(y) = z$ is described as follows:

$$a = \text{ReLU}(yW_a), \qquad\qquad W_a \in \mathbb{R}^{d\times d},$$
$$b = aW_b \qquad\qquad\qquad W_b \in \mathbb{R}^{d\times d}, \qquad (4)$$
$$z = \gamma_2 \odot \frac{y+b-\mu_{y+b}}{\sigma_{y+b}} + \beta_2, \qquad \gamma_2, \beta_2 \in \mathbb{R}^d,$$

where the parameter $w$ consists of MLP weights $(W_a, W_b)$, Layer-Norm weights $(\gamma_2, \beta_2)$. Some network architectures may be different. However, this does not affect the authorization because the authorization process mainly relies on $w_b$, which is a universal structure.

***Authorization Protocol.*** Let $\pi_{i+1} \in \{0, 1\}^{d\times d}$ denote a permutation matrix of the next attention block. We permute the parameters $w$ as follows:

$$W'_a = \pi_i^T W_a, \quad W'_b = \pi_{i+1}^T W_b, \quad \gamma'_2 = \gamma_2 \pi_{i+1}, \quad \beta'_2 = \beta_2 \pi_{i+1}. \quad (5)$$

With the permuted weights (denoted as $w'$), taking $y\pi_i$ (output of previous permuted attention block) as input, the first linear layer of the permuted MLP block can be described as follows:

$$a' = \text{ReLU}(y\pi_i \pi_i^T W_a) = a. \qquad (6)$$

To enable authorization to occur in an encrypted state, a random mask $m$ (just as the OTP) is introduced in TEE. Meanwhile, TransLinkGuard introduces $\pi_{i+1}$ to conceal $m$ (otherwise, $m$ could be discerned from the difference between $a'$ and $a'+m$). The computation carried out by TEE is as follows:

$$a'' = a'\pi_{i+1} + m_1\pi_{i+1} = a\pi_{i+1} + m\pi_{i+1}, \qquad (7)$$

where $a''$ is the encrypted feature, meaning that even for the same $a'$, the value of $a''$ produced is different, which protects the mapping from plaintext ($a'$) to encrypted state ($a''$) from being cracked.

To reduce the computational load executed within TEE, the computations with $W'_a$ are offloaded to GPUs:

$$b'' = a''\pi_{i+1}^T W_b = (a\pi_{i+1} + m\pi_{i+1})\pi_{i+1}^T W_b = b + mW_b, \qquad (8)$$

where $a''$ remains encrypted (by $mW_b$).

The second step of authorization consists of two parts: decryption (eliminate $mW_b$) and authorization (introduce $\pi_{i+1}$). We ensure the security of this process at both the hardware and algorithmic. From the algorithmic level, the attacker does not know the conversion relationship from encrypted state to plaintext, it effectively conceals $\pi_{i+1}$. From the hardware level, to protect the authorization process from runtime attacks, we execute it within TEE:

$$b' = (b'' - mW_b)\pi_{i+1} = b\pi_{i+1},$$
$$y'' = y'\pi_i^T \pi_{i+1} = y\pi_{i+1}, \qquad\qquad (9)$$
$$z' = \gamma_2\pi_{i+1} \odot \frac{b\pi_{i+1} + y\pi_{i+1} - \mu_{y+b}}{\sigma_{y+b}} + \beta_2\pi_{i+1} = z\pi_{i+1}.$$

Note that following prior work [55], computing $mW_b$ can be conducted by the model provider or inside TEE in an offline phase. Both strategies do not increase the overhead of online inference or impede its efficiency [64].

In conclusion, the permuted transformer layer can be represented as $g_{w'}(f_{\theta'}(x\pi_i), \pi_{i+1}) = z' = z\pi_{i+1}$, where $\pi_i$ originates from the authorization of the previous layer and $\pi_{i+1}$ is introduced by TEE to authorize the next transformer layer. In particular, for a transformer model with $n$ transformer layers. The first permutation matrix $\pi_1$ and the last permutation matrix $\pi_{n+1}$ are both equal to identity matrix $I$, thereby enabling the correct inference.

***Security Analysis.*** Potential attackers might attempt to steal the locked model by the recovery of permuted parameters. However, it is impossible as the probability of guessing the correct $\pi$ is $1/(d!)$ for each transformer layer. In practice, $d$ is typically larger than 512, e.g., $d = 4096$ in LLaMA [54].

Another strategy for stealing the locked model is to crack (or approximate) the authorization process based on its functional behavior. Notably, TransLinkGuard employs the OTP to make any attempt at approximating the authorization process unfeasible. This is because, even with identical inputs, the TEE produces different outputs for each inference.

However, a sophisticated attacker might attempt to approximate the authorization process on a larger scale, attempting to map the relationship from the start ($u'$) to the end ($z\pi_{i+1}$) to circumvent the OTP encryption. Nonetheless, it is also impractical (proved in Section 4.4), as the second linear layer of the MLP block is involved in the authorization process, requiring the attacker to approximate a substantial portion of the parameters—about a third of the total network parameters [10].

## 4 EXPERIMENTS

In this section, we perform extensive experiments to answer the following research questions:

> **RQ1:** How does TransLinkGuard compare with other representative defenses in security? **RQ2:** How does TransLinkGuard's efficiency compare to other defenses? **RQ3:** Does TransLinkGuard sacrifice the accuracy of the model?

### 4.1 Evaluation Settings

***Datasets.*** To evaluate TransLinkGuard's adaptability and effectiveness in varied real-world contexts, we select various from

| | | No-Shield | Serdab | SOTER | ShadowNet | DarkneTZ | OLG | **Ours** | Black-box |
|---|---|---|---|---|---|---|---|---|---|
| RoBERTa | SQuAD | 81.66% | 51.05% | 47.54% | 81.63% | 35.91% | 67.55% | 4.24% | 2.75% |
| | MNLI | 87.77% | 82.98% | 75.53% | 88.01% | 73.73% | 77.17% | 32.82% | 41.69% |
| | QQP | 91.14% | 81.51% | 85.55% | 91.24% | 90.94% | 85.33% | 66.98% | 69.33% |
| | SST-2 | 94.03% | 84.63% | 78.55% | 93.72% | 92.89% | 80.96% | 70.76% | 75.80% |
| BART | SQuAD | 78.28% | 63.71% | 67.92% | 68.01% | 44.91% | 62.05% | 3.98% | 4.91% |
| | MNLI | 84.10% | 84.81% | 80.41% | 84.14% | 81.84% | 82.89% | 40.51% | 41.71% |
| | QQP | 91.47% | 78.82% | 83.81% | 91.64% | 88.71% | 83.26% | 70.17% | 68.92% |
| | SST-2 | 93.10% | 88.42% | 82.91% | 82.47% | 90.48% | 87.61% | 75.30% | 73.62% |
| GPT-2 | SQuAD | 55.60% | 47.81% | 58.71% | 35.56% | 33.10% | 45.89% | 3.91% | 5.81% |
| | MNLI | 81.04% | 70.81% | 57.91% | 61.03% | 78.85% | 62.91% | 47.81% | 35.17% |
| | QQP | 88.55% | 71.14% | 72.06% | 88.62% | 81.41% | 70.75% | 74.52% | 70.59% |
| | SST-2 | 91.63% | 74.58% | 78.91% | 82.18% | 85.84% | 75.55% | 58.91% | 57.47% |
| ChatGLM-6B | GSM8k | 34.91% | 12.81% | 15.26% | 34.85% | 28.24% | 13.35% | 4.91% | 3.89% |
| | Spider | 19.24% | 5.92% | 12.30% | 17.81% | 12.61% | 6.67% | 4.29% | 5.13% |
| | PubMedQA | 69.50% | 14.00% | 5.00% | 48.00% | 7.00% | 16.50% | 0.00% | 1.00% |
| | SQuAD | 76.00% | 43.34% | 35.25% | 55.23% | 16.28% | 45.18% | 10.82% | 7.81% |
| LLaMA2-7B | GSM8k | 42.68% | 15.92% | 12.60% | 42.12% | 3.15% | 14.89% | 0.47% | 1.04% |
| | Spider | 35.81% | 8.91% | 6.47% | 14.52% | 5.83% | 10.82% | 4.50% | 3.15% |
| | PubMedQA | 71.00% | 13.50% | 14.00% | 49.50% | 17.00% | 12.50% | 0.00% | 0.00% |
| | SQuAD | 68.34% | 45.01% | 25.91% | 69.03% | 26.34% | 33.90% | 6.91% | 4.51% |
| | Average | 2.50× | 1.81× | 1.73× | 2.23× | 1.73× | 1.80× | 1.01× | 1.00× |

**Table 2: Attack accuracy regarding representative defense schemes. The last row reports the average accuracy of each defense relative to the baseline black-box solutions. For each setting, we mark the lowest attack accuracy in yellow. Attack accuracy toward TransLinkGuard is marked with green.**

different domains. We assess models on the most representative sub-tasks of the standard GLUE benchmark [58] (SST-2, MNLI, QQP) and four distinct domain-specific datasets: GSM8k (mathematics) [9], Spider (code generation) [62], PubMedQA (medical question answering) [23], and SQuAD (reading comprehension) [44].

**Models.** We focus on several commonly used representative transformer models for validation, including three medium-sized models: RoBERTa (encoder-only) [32], BART (encoder-decoder) [28], GPT-2 (decoder-only) [43], and two large models, LLaMA2-7B [54] and ChatGLM-6B [10], to encompass models of different architectures and scales. We equip BART, GPT-2, and RoBERTa with classification heads for text classification tasks and consistently designed prompts for effective training for generative tasks.

**Metric.** For performance evaluation, accuracy is uniformly used as the metric. For classification tasks, correct category output is considered accurate, while for generation tasks like GSM8k and Spider, precise matching of the answer in the output is deemed correct. For security, we use model-stealing accuracy (denoted as "MS acc"). Higher MS acc indicates better effectiveness of MS, i.e., poorer security of the defense. To measure the efficiency cost of models, we follow prior work to use Floating Point Operations (FLOPs) as the efficiency cost metric [17, 52]. FLOPs is a platform-irrelevant metric used to assess efficiency costs by counting the total number of multiplication and addition operations conducted inside TEEs. For clarity, we define %FLOPs as the ratio of FLOPs over the total FLOPs of the model.

**Implementation Details.** We conduct our experiments using the *Huggingface transformers library*[1]. For optimization, we use the AdamW optimizer and a linear learning rate scheduler with

an initial rate of 5e-5. Our reported results are based on the runs that achieved the highest performance, consistent with real-world practices prioritizing optimal model performance.

## 4.2 Comparisons

**Representative Defenses.** For existing PTSE solutions, we select four representative solutions for comparison. (1) SOTER [49]: we chose SOTER as it demonstrated the best performance against MS attack in the evaluations conducted by [64] in existing PTSE solutions. (2) Serdab [11]: we selecte Serdab as it focuses on protecting the shallow layers of networks, which are often more crucial for transformer models [61]. (3) DarkenTZ [36]: we chose DarkneTZ because, according to prior work [16, 35], it is the state-of-the-art (SOTA) solution for protecting edge models. (4) ShadowNet [51]: we chose ShadowNet as it is the most recently published work with significant influence in the field.

**Baselines.** To ease the comparison, we also provide baseline evaluation results. (1) No-shield: we consider the white-box solution as the easiest baseline because the adversary can directly use the offloaded $M_{vic}$ as $M_{sur}$ and does not need to train the model. (2) Black-box: we consider a black-box setting, where attackers can only identify the $M_{vic}$'s architecture. (3) One-layer-Guard (referred to as OLG): this is a variant of TransLinkGuard that only permutes the first transformer layer and manually authorizes it. We use this variant to assess security when only a single layer is permuted.

## 4.3 Configuration Settings

Current PTSE methods are fundamentally designed for CNNs. To adapt them for use with transformer models, we rigorously configure each PTSE solution based on its papers. Specifically, for SOTER, TEE shields 20% randomly selected layers and multiplies the other layers with a scalar to conceal the weight values. For Serdab, the

[1]https://huggingface.co/docs/transformers/index

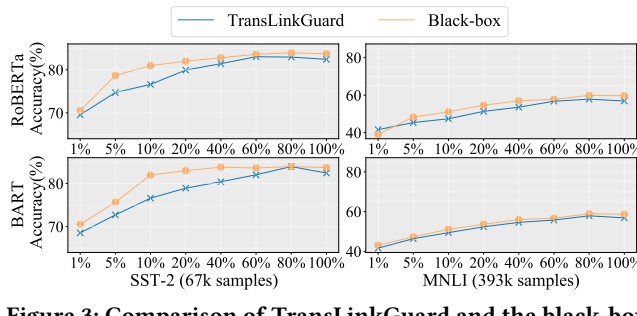

**Figure 3: Comparison of TransLinkGuard and the black-box protection against MS attacks with different sizes of dataset.**

|  |  | No-Shield | **TransLinkGuard** | Black-box |
|---|---|---|---|---|
| RoBERTa | SST-2 | 94.03% | 78.12% | 75.80% |
|  | MNLI | 87.77% | 45.40% | 41.69% |
|  | QQP | 91.14% | 69.75% | 69.33% |
| LLaMA2 | SQuAD | 68.34% | 4.91% | 4.51% |
|  | Spider | 35.81% | 6.18% | 3.15% |
|  | GSM8k | 42.68% | 5.29% | 1.04% |

**Table 3: Security evaluation of TransLinkGuard against authorization process simulation attack.**

TEE shields the first transformer layers. ShadowNet obfuscates and offloads all the linear transformation layers with matrix transformation and filter permutation, and we follow the prior work [64] to use the decoded weights to initialize $M_{init}$. For DarkneTZ, the last transformer layer and subsequent parts are put into TEE.

## 4.4 Security Guarantee

*Security Guarantee.* In this subsection, we assess if the defense is sufficiently secure against potential MS attacks. Specifically, we consider a realistic scenario in which attackers have a small amount of data (such as 1% of the training dataset) and attempt to steal the $M_{vic}$ by MS attack. The attack pipeline is the same as in Section 2.3.

Table 2 reports the results: in all cases, the attack accuracies of TransLinkGuard (marked with green) are comparable with black-box protection and are better than the best of existing defenses (marked with yellow). Specifically, the relative accuracy of TransLinkGuard compared to the black-box baseline is 1.01×, while the relative value of the best defense, Serdab, is 1.81×. Notably, the relative accuracy of OLG (1.80×) is similar to that of Serdab (1.81×), which achieves protection by placing the first layer into the TEE (i.e., a black-box protection). This implies that even with protection applied to a single transformer layer, the permutation strategy of TransLinkGuard achieves black-box-level security.

*Security under Other Assumptions of Data.* In *security guarantee*, we evaluate a realistic adversary with a small amount of training data. Although our assumption of the adversary is realistic [19, 45, 60, 64], we still evaluate the security of TransLinkGuard with an ideal adversary with a large amount of data to verify whether TransLinkGuard ensures the security of models under extreme conditions. Figure 3 shows accuracies between our approach and black-box protection on various data sizes. In all cases, the attack accuracies are lower than or close to the black-box baseline. To summarize, under a different assumption of training data,

TransLinkGuard demonstrates robust security for models even when faced with an ideal adversary equipped with a large dataset.

*Security against Sophisticated Attackers.* In this subsection, we assess the security of TransLinkGuard against attackers who are familiar with the principles of TransLinkGuard and implement attacks accordingly. Specifically, the core mechanism of authorization involves a row-wise permutation of features between the second linear layer and the norm layer within each MLP block. This authorization process is initially envisioned as a two-step multiplication (first by $M_v$ and then by $\pi_{i+1}$). However, it is achievable through a single operation where $M_v$ and $\pi_{i+1}$ are multiplied to result in $M_v\pi_{i+1}$. Expanding on this, the attackers copy and freeze all other components, then reinitialize and train both the $M_v$ and the norm layer to bypass TEE's authorization (refer to as *authorization process simulation* attack).

The results are compiled in Table 3; the attack accuracy is similar between TransLinkGuard and the black-box baseline but significantly lower than the no-shield baseline. We believe the outstanding defense effectiveness is due to the massive parameters of $M_v$, which makes it difficult for attackers to simulate. Specifically, the number of parameters they need to simulate is about one-third of the entire network, which is much higher than the existing PTSE methods (where the highest, SOTER, protects about 10% of the parameters within an acceptable efficiency cost).

> **Answer to RQ1:** TransLinkGuard surpasses other representative defenses in security and achieves **black-box-level** security guarantees to a **single transformer layer**. Furthermore, TransLinkGuard consistently achieves black-box-level security under various attack assumptions.

## 4.5 Efficiency Cost

To answer **RQ2**, we quantitatively compare TransLinkGuard with the *%FLOPs* of other defenses in Table 4. Taking an example length of 128 as input, we calculate the overhead of a single inference. TransLinkGuard achieves a similar *%FLOPs* than other defenses. Specifically, the additional overhead caused by TransLinkGuard is less than 0.1% for all cases. The computational overhead for protection at a single layer (OLG column) executed in TEE is minimal (all less than 0.01%), which allows the protection to remain negligible even when extended to all transformer layers (TransLinkGuard column). On the contrary, the efficiency cost of other defenses ranges from 3.0337% to 38.0071%. That is, TransLinkGuard takes 30× less efficiency cost to achieve the highest (black-box) security.

> **Answer to RQ2:** The overhead of TransLinkGuard is negligible. The efficiency cost of TransLinkGuard is 30× less than other PTSE solutions.

## 4.6 Accuracy Loss

To answer this research question, we compare the accuracy between the original model $M_{ori}$ and the derived model $M_{vic}$. The result is shown in Table 5. In general, TransLinkGuard does not lead to a noticeable loss of accuracy. Consistent with formulaic derivation, there is no difference in accuracy between $M_{ori}$ and $M_{vic}$ in most cases. However, for some specific cases, accuracy slightly fluctuates (marked in blue). For example, with RoBERTa on SST-2, there is a

| Models | Original *FLOPs* | Additional *FLOPs*(%*FLOPs*) in TEEs | | | | | |
| | | Serdab | SOTER | ShadowNet | DarkneTZ | OLG | TransLinkGuard |
|---|---|---|---|---|---|---|---|
| RoBERTa | 2.236E+10 | 1.8633E+09 (8.3422%) | 4.486E+09 (20.0615%) | 7.793E+09 (30.0983%) | 1.863E+09 (8.3422%) | 1.278E+06 (**0.0057%**) | 1.534E+07 (**0.0686%**) |
| BART | 2.539E+10 | 2.1158E+09 (8.4151%) | 5.209E+09 (20.5164%) | 9.649E+09 (38.0071%) | 2.116E+09 (8.4151%) | 1.278E+06 (**0.0050%**) | 1.534E+07 (**0.0604%**) |
| GPT-2 | 2.236E+10 | 1.8633E+09 (8.3422%) | 4.445E+09 (19.8792%) | 7.793E+09 (30.0983%) | 1.863E+09 (8.3422%) | 1.278E+06 (**0.0057%**) | 1.534E+07 (**0.0686%**) |
| ChatGLM-6B | 1.598E+12 | 5.510E+10 (3.4482%) | 3.119E+11 (19.5154%) | 5.101E+11 (31.9212%) | 5.537E+10 (3.4650%) | 6.816E+06 (**0.0004%**) | 1.908E+08 (**0.0119%**) |
| LLaMA2-7B | 1.700E+12 | 5.1573E+10 (3.0337%) | 3.641E+11 (21.4197%) | 4.566E+11 (26.8588%) | 5.170E+10 (3.0414%) | 6.127E+06 (**0.0004%**) | 1.961E+08 (**0.0115%**) |

**Table 4: The results of additional inference overhead. The table includes the original model's FLOPs ("Original FLOPs"), the additional overhead in TEE, and its proportion to the original model's FLOPs.**

| | SQuAD | SST-2 | MNLI | QQP | GSM8k | Spider | PubMedQA |
|---|---|---|---|---|---|---|---|
| RoBERTa | 81.66%/81.66% | 94.03%/94.01% | 87.77%/87.78% | 91.14%/91.14% | - | - | - |
| BART | 78.28%/78.28% | 93.10%/93.10% | 84.10%/84.10% | 91.47%/91.47% | - | - | - |
| GPT-2 | 55.60%/55.58% | 91.63%/91.63% | 81.04%/81.04% | 88.55%/88.55% | - | - | - |
| ChatGLM-6B | 76.00%/76.00% | - | - | - | 34.91%/34.91% | 19.24%/19.24% | 69.50%/69.50% |
| LLaMA2-7B | 68.34%/68.34% | - | - | - | 42.68%/42.68% | 35.81%/35.81% | 71.00%/71.00% |

**Table 5: The accuracy comparison between the original model ($M_{ori}$) and the protected model ($M_{vic}$). The accuracy is presented in the form of $M_{ori}/M_{vic}$. Cells showing changes in accuracy are highlighted in blue.**

minor decrease of 0.02% in accuracy. Interestingly, despite these fluctuations, we observe an improvement of 0.01% on the MNLI. Therefore, we consider that the minor accuracy fluctuations are caused by data precision limitations rather than by the defense itself, which is inevitable.

> **Answer to RQ3:** While significantly outperforming existing defenses in terms of both security and efficiency, TransLinkGuard maintains the model's accuracy without compromise.

## 5  LIMITATION AND DISCUSSION

***Runtime Efficiency.*** Although FLOPs, as a platform-irrelevant metric, demonstrate that TransLinkGuard incurs minimal additional overhead, the diversity of hardware and variations in testing environments prevent us from systematically evaluating the actual overhead. Future research could conduct extensive performance tests across various hardware platforms and environments to ensure a comprehensive efficiency analysis.

***More Models and Tasks.*** This work demonstrates that TransLinkGuard is exceptionally effective in the most commonly used tasks, such as text generation, text classification, and reading comprehension. However, other tasks, such as relation extraction and language modeling, remain to be evaluated in further investigation.

## 6  OTHER RELATED WORK

***TEE in GPUs.*** Recent work explored implementing trusted architectures directly inside GPUs to achieve black-box protection [18, 38, 57]. Such solutions require customizing hardware and are designed for server centers. However, our solution is primarily for users' end devices, where it is impractical and costly for model providers to modify hardware or ship firmware. Thus, this paper employs commercial GPUs as a generic solution.

***Whole Model Execution by TEE.*** In addition to PTSE solutions, there are also existing works exploring the placement of entire

models into TEEs [15, 26, 27, 29, 50]. However, these works have significant limitations as they often unacceptably sacrifice the efficiency of the protected models.

***Privacy-centric Weights Protection.*** Privacy-centric weight protection strategies [64] specifically train privacy-related data onto additional parameters and only target these privacy-centric weights for protection. However, this strategy is unsuitable for intellectual property protection as it focuses solely on privacy-sensitive parts, thus neglecting other critical parameters that are equally important for the functionality of the model. This is precisely the main goal that TransLinkGuard addresses.

***Secure Computation Methods.*** Prior secure computation approaches use either homomorphic encryption (HE) [13] or multiparty computation (MPC) [24, 46, 63]. However, HE-based techniques are orders of magnitude slower than the state-of-the-art (nonsecure) model inference. MPC-based approaches involve multiple participants requiring network connectivity, which is unsuitable for real-time tasks or offline usage.

## 7  CONCLUSIONS

In this paper, we introduce a protection method named TransLinkGuard for edge-deployed LLMs. Unlike existing methods, we utilize a request-level authorization mechanism to safeguard these models. Importantly, through this authorization mechanism, TransLinkGuard achieves comprehensive protection throughout the entire model edge deployment process (before and during the inference stage). The derivation of formulas and experiments across various tasks demonstrate that only with appropriate authorization the TransLinkGuard-protect model can operate normally. Furthermore, comprehensive experiments indicate that TransLinkGuard exhibits exceptional security and efficiency compared to the existing PTSE approaches. In conclusion, TransLinkGuard is a solution for the edge deployment of proprietary LLMs, providing model owners with the means to safeguard their valuable intellectual property.

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
