# OpenReview forum: "TransLinkGuard: Safeguarding Transformer Models Against Model Stealing in Edge Deployment"
_acmmm.org/ACMMM/2024/Conference — MM2024 Poster_

### Official Review · Reviewer_2Lbf · 2024-05-15

**Rating:** 4
**Confidence:** 2

**Summary:**

The paper introduces TransLinkGuard, an innovative model protection approach designed to safeguard large language models (LLMs) against model stealing attacks when deployed on edge devices.  It is a lightweight authorization module housed within a secure execution environment, such as a Trusted Execution Environment (TEE). Its main innovation lies in rearranging the weight matrix of linear layers within the model, ensuring that only inputs permuted according to the authorization module's order can be correctly processed. This paper shows that TransLinkGuard outperforms existing PTSE approaches with evaluation experiments in terms of security guarantee and efficiency cost.

**Strengths:**

TransLinkGuard works as a robust solution for protecting valuable transformer models from theft while maintaining their usability and performance in edge computing environments. It employs a lightweight authorization module that can authorize each request individually based on its input. This provides a high level of control and security. The authorization module operates within a trusted execution environment (TEE), which is an isolated hardware enclave. The paper provides diverse transformer model evaluation experiments which ensures that the protection method is not limited to specific model architectures but is broadly applicable. And expeiment results show that TransLinkGuard performs better than existing PTSE defenses in security and efficiency.

**Limitations:**

1. While the paper effectively presents TransLinkGuard as a solution for safeguarding large language models (LLMs) in edge deployment scenarios, it may not fully account for the increased hardware costs associated with its implementation. Edge devices are typically resource-constrained, requiring solutions that are lightweight and efficient. TransLinkGuard's reliance on a Trusted Execution Environment (TEE) and the incorporation of encryption for secure authorization introduces additional hardware security requirements. The paper could benefit from a discussion on the quantifiable costs of integrating TEE and the computational overhead associated with cryptographic operations, especially considering the budget-sensitive nature of many edge computing applications. A cost-benefit analysis comparing the security enhancements provided by TransLinkGuard against the potential increase in hardware requirements and costs would be valuable for practitioners considering its adoption.
2. While TransLinkGuard leverages Trusted Execution Environments (TEEs) to provide a secure computation framework for protecting large language models during edge deployment, the paper does not extensively discuss the potential new threats that TEEs themselves might introduce. TEEs, while offering a layer of hardware-based security, can also become a target for sophisticated attacks such as those exploiting side-channel vulnerabilities or hardware backdoors. The paper could benefit from an exploration of these risks and how they might be mitigated.
3. While the paper highlights the superior performance of TransLinkGuard over SOTER, a recent Partial Trusted Execution Environment (PTSE) solution, in terms of accuracy and security, it does not provide a detailed analysis to explain the performance discrepancy. The paper states that SOTER demonstrated less effective accuracy, but it does not explore the underlying reasons for this outcome. A thorough investigation into why TransLinkGuard outperforms SOTER would be beneficial for the reader.

**Suitability:**

2

---

### Official Review · Reviewer_Remp · 2024-05-22

**Rating:** 3
**Confidence:** 3

**Summary:**

The paper introduces TransLinkGuard, a novel method for protecting LLMs by employing a locked model at the edge, coupled with an authorization module within a secure environment. This setup achieves request-level authorization, enhancing security by ensuring that only authorized inputs are processed correctly. It leverages a lightweight row permutation strategy to maintain the efficiency and security of the model without compromising its accuracy.

**Strengths:**

-	Comprehensive Security Measures: TransLinkGuard offers a robust solution for safeguarding the intellectual property of models. It effectively addresses threats like model stealing and runtime attacks, providing a holistic security framework.
-	Enhanced Security with Request-Level Authorization: The implementation of request-level authorization ensures that even if an attacker gains access to the locked model, it remains unusable without the corresponding authorization module. This significantly strengthens the security posture.
-	Efficient Authorization Mechanism: The use of a row permutation strategy minimizes the overhead on the authorization module, maintaining high efficiency and low latency in model operations.
-	Preservation of Model Accuracy: The experiments demonstrate that the accuracy of the TransLinkGuard-protected model matches that of the original, affirming that the security enhancements do not degrade performance.

**Limitations:**

-	Defensive capability of the method: In fact, there have been recent works targeting attacks on channel permutations, such as "No Privacy Left Outside: On the (In-)Security of TEE-Shielded DNN Partition for On-Device ML" [1]. The defensive capability of this paper against such attacks remains to be verified.
-	Novelty: The core idea of this paper lies in the permutation of linear channels. However, in previous works such as "ShadowNet: A Secure and Efficient On-device Model Inference System for Convolutional Neural Networks" [2], a defense mechanism for permutation/shuffle of the network structure was explicitly proposed. The authors should give credit to these works in the introduction or related works section and elaborate on how this paper differs from those works.
-	Writing and formatting issues: This paper has some obvious formatting issues: for example, the first three references in the first introduction paragraph of the paper are incorrect; the table caption is misplaced and should be positioned above the table, etc.

[1] No Privacy Left Outside: On the (In-)Security of TEE-Shielded DNN Partition for On-Device ML. https://arxiv.org/abs/2310.07152

[2] ShadowNet: A Secure and Efficient On-device Model Inference System for Convolutional Neural Networks. https://arxiv.org/abs/2011.05905

**Suitability:**

2

---

### Official Review · Reviewer_rqeA · 2024-05-24

**Rating:** 4
**Confidence:** 3

**Summary:**

The paper presents a new plug-and-play model protection approach called TransLinkGuard to address the issue of model stealing on edge devices. The proposed method effectively resolves the limitations of existing defense mechanisms in simultaneously satisfying four critical protection properties. By deploying a lightweight authorization module within a secure environment, such as a Trusted Execution Environment (TEE), TransLinkGuard achieves real-time authorization for each request, ensuring security with minimal runtime overhead. Experimental results demonstrate that TransLinkGuard outperforms existing PTSE methods by providing enhanced security guarantees, reduced overhead, and no/low loss in accuracy.

**Strengths:**

- This paper is well-written.

- The evaluation is comprehensive and results are convincing.

- Compared to existing PTSE methods, the proposed TransLinkGuard offers enhanced security protection (i.e., protect all Transformer layers) , while concurrently reducing computational overhead without any compromise in accuracy.

**Limitations:**

- While the experimental results and methodology suggest that the proposed method is applicable to all transformer models, it is important to note that the author's motivation primarily focuses on LLMs. Therefore, it would be beneficial for the authors to address this limitation and provide further justification for the generalizability of their approach to a broader range of transformer models.

- The authors did not mention the experimental platform, making it difficult to ascertain how the TEE component was implemented and evaluated. Additionally, it is unclear whether this aligns with a real-world edge deployment scenario.

**Suitability:**

2

---

### Official Review · Reviewer_ZW3C · 2024-05-26

**Rating:** 5
**Confidence:** 3

**Summary:**

In this paper, the authors propose a plug-and-play model protection approach, TransLinkGuard, against model stealing attacks on edge devices. This approach utilizes a lightweight authorization module within a Trusted Execution Environment (TEE) to ensure that each model request is dynamically authorized based on its input, maintaining protection even if the model is copied. Specifically, the authorization is built upon a novel permutation strategy that manipulates the weights of each transformer layer in the original model. The evaluation results demonstrate that TransLinkGuard provides robust security akin to black-box models while introducing negligible computational overhead, making it highly effective and practical for real-world applications.

**Strengths:**

- TransLinkGuard incorporates a novel permutation strategy coupled with the use of One Time Pad (OTP) to encrypt the authorization process. This approach ensures that even if a model is physically copied, unauthorized users cannot execute successful model-stealing attacks. This proactive security measure significantly surpasses traditional methods that typically rely on passive protection or fail to secure copied or duplicated models effectively

- TransLinkGuard also ensures computational efficiency. The authorization module is designed to be lightweight and integrated seamlessly with the model's operational flow, ensuring that the protection mechanism does not introduce significant latency or computational cost. This aspect is crucial for edge devices that have limited computation resources.

- The authors conduct extensive experiments to evaluate the performance of TransLinkGuard, covering a range of scenarios and transformer models. The evaluation results showcase the effectiveness of TransLinkGuard across different settings. The real-world applicability of this method is evident from its performance, which matches the security level of black-box models without their associated drawbacks.

**Limitations:**

- The paper introduces TransLinkGuard, which employs a permutation strategy to protect model parameters. Still, it does not discuss whether the permutation matrix is fixed across devices or uniquely generated per device. If the matrix is indeed fixed and shared, a single compromise could potentially lead to widespread security breaches across all devices using TransLinkGuard. This situation is analogous to the vulnerabilities inherent in symmetric encryption, where the exposure of a single key can compromise the security of the entire system. An in-depth analysis of this aspect and potential mitigation strategies would be beneficial.

- TransLinkGuard relies on the robust security features of TEEs to protect the execution of its critical authorization processes; however, TEEs themselves are not immune to sophisticated attacks, especially side-channel attacks that could potentially expose sensitive data such as the permutation matrix or cryptographic keys. The paper could benefit from a more detailed examination of these potential vulnerabilities, including recent advances in side-channel attack methodologies and their implications for the security of TransLinkGuard.

- The TEE is a core component of the security mechanism in TransLinkGuard, but the paper lacks detailed information on how the TEE is implemented and integrated within the overall system. Specific details such as the type of TEE used (TrustZone or SGX), the configuration of the TEE environment, and how it interacts with the transformer models and authorization modules are crucial for replicating and validating the security claims made. Moreover, the absence of implementation details might hinder the ability of other researchers or practitioners to accurately assess the feasibility, performance, and security of the TEE component in real-world deployments.

**Suitability:**

3

---

### Meta-Review · Area_Chair_nEjS · 2024-07-01

**Recommendation:** Accept (Poster)
**Confidence:** 4

**Metareview:**

Summary:

This paper proposes TransLinkGuard, a plug-and-play model protection approach against model stealing attacks on edge devices. Utilizing a lightweight authorization module within a Trusted Execution Environment (TEE), TransLinkGuard dynamically authorizes each model request based on its input, maintaining protection even if the model is copied. The novel permutation strategy manipulates the weights of each transformer layer, providing robust security with minimal computational overhead.

Strengths:
1. TransLinkGuard's novel permutation strategy and the use of One Time Pad (OTP) encryption ensure robust protection against model-stealing attacks.
2. The lightweight authorization module is designed to integrate seamlessly with the model's operational flow, ensuring minimal latency and computational cost.
3. Extensive experiments demonstrate TransLinkGuard's effectiveness across various transformer models and scenarios, matching the security level of black-box models without their associated drawbacks.

Limitations:
1. The paper does not discuss whether the permutation matrix is fixed across devices or uniquely generated per device, which could lead to security vulnerabilities similar to symmetric encryption.
2. While TransLinkGuard relies on TEEs for security, the paper does not address potential side-channel attacks or other sophisticated threats that could compromise TEEs.
3. The paper lacks specific information on how the TEE is implemented and integrated, making it difficult for other researchers to replicate and validate the security claims.
4. The paper primarily focuses on LLMs, and it needs to provide further justification for the generalizability of TransLinkGuard to a broader range of transformer models.

According to the final ratings, all reviewers tend to accept this submission. Therefore, I will recommend this paper as Accept (Poster).